# A Patient-Centered Conceptual Model of AYA Cancer Survivorship Care Informed by a Qualitative Interview Study

**DOI:** 10.3390/cancers16173073

**Published:** 2024-09-04

**Authors:** Marlaine S. Figueroa Gray, Lily Shapiro, Caitlin N. Dorsey, Sarah Randall, Mallory Casperson, Neetu Chawla, Brad Zebrack, Monica M. Fujii, Erin E. Hahn, Theresa H. M. Keegan, Anne C. Kirchhoff, Lawrence H. Kushi, Hazel B. Nichols, Karen J. Wernli, Candice A. M. Sauder, Jessica Chubak

**Affiliations:** 1Kaiser Permanente Washington Health Research Institute, 1730 Minor Avenue Suite 1600, Seattle, WA 98101, USA; lily.x.shapiro@kp.org (L.S.); caitlin@processpioneersllc.com (C.N.D.);; 2Cactus Cancer Society, 2323 Broadway, Oakland, CA 94612, USA; 3Veterans Affairs Greater Los Angeles Healthcare System, Center for the Study of Healthcare Innovation, Implementation & Policy, 16111 Plummer Street (152), North Hills, CA 91343, USA; 4School of Social Work, University of Michigan, 1080 S. University, Ann Arbor, MI 48109, USA; zebrack@umich.edu; 5Department of Research and Evaluation, Kaiser Permanente Southern California, 100 S. Los Robles Avenue, Pasadena, CA 91101, USA; erin.e.hahn@kp.org; 6Division of Hematology and Oncology, University of California Davis Comprehensive Cancer Center, 4501 X Street Suite 3016, Sacramento, CA 95817, USA; tkeegan@ucdavis.edu; 7Huntsman Cancer Institute, University of Utah, 2000 Circle of Hope, Office 4715, Salt Lake City, UT 84112, USA; 8Division of Research, Kaiser Permanente Northern California, 2000 Broadway, Oakland, CA 94612, USA; 9Gillings School of Global Public Health, University of North Carolina, 2104F Mcgavran-Greenberg Hall CB #7435, Chapel Hill, NC 27599, USA; 10Division of Surgical Oncology, Department of Surgery, University of California Davis Comprehensive Cancer Center, 4501 X Street Suite 3010, Sacramento, CA 95817, USA

**Keywords:** AYA survivorship care, conceptual model, oncology, peer support, mental health, care coordination

## Abstract

**Simple Summary:**

Adolescents and young adults (AYAs) experiencing cancer have support needs that differ from older adult cancer patients. The aim of our qualitative study was to determine the holistic needs of AYA cancer survivors and to develop a patient-centered conceptual model of AYA survivorship care. Analysis of our interview results reveals eight key domains critical to holistic patient-centered AYA survivorship care. We offer a conceptual model that differs from current conceptual models of AYA survivorship care by centering the patient and their support systems, emphasizing the need for continuing supportive navigation, and the importance of repeated support along the identified domains over time.

**Abstract:**

**Purpose**: Conceptual models provide frameworks to illustrate relationships among patient-, provider-, system-, and community-level factors that inform care delivery and research. Existing models of cancer survivorship care focus largely on pediatric or adult populations whose needs differ from adolescents and young adults (AYAs). We developed a patient-centered conceptual model of AYA survivorship care. **Methods:** We conducted a narrative literature review of current conceptual and theoretical models of care. We engaged AYA cancer survivors (n = 25) in semi-structured one-hour telephone interviews. Most participants were in their 20s and 30s, and the majority (84%) were women. Recruitment was stratified by age and time since cancer diagnosis. We conducted a thematic analysis of interview transcripts to identify themes that exemplified patient-centered care. **Results**: Most participants identified as white and female. Leukemia and breast cancer were the most common cancer types. Main themes included the need for (1) care coordination, (2) ongoing mental health support, (3) connection to AYA peer support, (4) support during fertility preservation efforts, (5) support with financial burden, (6) support for quality of life, (7) information about and support with side effects and late effects, and (8) attention to the unique needs of young adults. **Conclusions:** We present a patient-centered conceptual model of AYA survivorship care needs that can inform future cancer care delivery and research.

## 1. Introduction

Adolescent and young adult (AYA) cancer survivors are in need of holistic survivorship care that meets their unique needs. Existing conceptual models of cancer survivorship care tend to focus on either pediatric populations or patients diagnosed over the age of 40 years, whose care experiences and needs often differ from those of AYAs who are diagnosed with cancer between ages 15 and 39 years [1]. The purpose of a conceptual model of care is to diagram the various factors that influence a person’s experience as they move through the healthcare system. Conceptual models provide an organized context for understanding key relationships among patient-, provider-, system-, and community-level factors that shape and inform cancer care delivery. Many existing models of cancer care, such as the Anderson model [2], reflect a biomedical orientation that emphasizes surveillance for cancer recurrence or new cancer primaries, as well as symptom management, emphasizing the post-treatment phase while neglecting important aspects of the AYA cancer experience.

In our narrative review of the literature, we focused on studies that adapted the Anderson [2] conceptual model of health care utilization or studies that developed their own conceptual model related to AYA cancer care and experience. Many of the prior AYA models focus on one particular aspect of care delivery—for instance, fertility concerns [3,4], sexual health [5], financial burden [6], physical exercise [7], career trajectory [8], or mental health [9]—including psychosocial adjustment [10], resilience [11], social networks [12], and fear of cancer recurrence [13]. Several models focus specifically on the need for continued surveillance and support for AYA cancer survivors after treatment [1,9,14,15].

A few models in our review were more broadly focused on multiple aspects of the cancer care experience. Taylor et al. [16] developed a conceptual model of cancer treatment experience based on published research. Fern et al. [17] offer a complementary framework to describe the lived experience throughout diagnosis and treatment based on interviews and highlight several key themes, including the life-changing impact of diagnosis, the importance of the delivery of healthcare information, and the necessity of peer and psychological support. Hammond [18] offers a contextual framework based on the diverse sociodemographic characteristics of AYAs and explores some of the social-environmental issues that are especially relevant to this age group, including precarious labor conditions, the salience of sexual orientation and gender identities, and the significance of cultural plurality.

Most of these models and frameworks move linearly through time; we did not find any that emphasized specifically the need for care touches at multiple timepoints in a more iterative or cyclical fashion, or the need for tailored care coordination support to meet the specific needs of this population. We, therefore, sought to address these gaps with a patient-centered conceptual model of AYA cancer survivorship care informed by interviews with AYA survivors about their cancer experience.

## 2. Methods

We employed a reflexive thematic analysis approach [19,20] to perform qualitative data collection and analysis [21]. The goal of our study was to elicit the lived experiences of AYAs who had experienced cancer in order to suggest how patient perspectives could be used to inform a patient-centered survivorship care model. We obtained institutional review board approval from KP-Northern California Region IRB. We report our methods in accordance with the consolidated criteria for reporting qualitative research (COREQ) [22].

### 2.1. Recruitment

We partnered with Cactus Cancer Society [23] (formerly Lacuna Loft), a support organization for AYAs, to purposively [24] recruit participants for this study from June through November 2021, given their wide reach in the AYA community and connections with other AYA organizations. Eligible participants were aged 18 years or older, were diagnosed with cancer between age 15 and 39 years [25], and resided in the USA (Figure 1).

The Cactus Cancer Society (CCS) introduced the study to their network via their email lists, blogs, and social media channels, which reach over 7500 young adults who have experienced cancer, including members of Black and White Cancer Survivors Foundation [26], Teen Cancer America [27], Young Adult Survivors United [28], Testicular Cancer Foundation [29], Sisters Network [30], and Imerman Angels [31]. The recruitment materials directed participants to an online eligibility screening survey. Prioritizing the perspectives of people from racially marginalized groups and those underrepresented in research, we contacted eligible participants to schedule interviews until we had recruited our target number (n = 30), though several of these interviews were unusable for analysis (n = 8) (see limitations). We conducted additional interviews (n = 3) for a total of 33 interviews conducted and 25 transcripts suitable for analysis. This number was adequate to provide thematic saturation and include diverse viewpoints [32,33]. Informed consent was obtained before beginning the interview. After completion of the interview, participants were sent a USD 40 gift card.

### 2.2. Interview Approach

Two team members (CD and MFG) conducted semi-structured, one-hour telephone interviews with the AYA cancer survivors. Interviews were audio recorded and professionally transcribed. Our interview guide was designed by a PhD-level medical anthropologist (MFG) to explore key constructs related to AYA survivorship. The guide format was based on an open-ended interview approach that explores the context of actions and feelings over time [34]. The guide content was informed by our narrative literature review of existing conceptual models of AYA survivorship care and the socioecological model of disease [35] and included domains representing individual, interpersonal and system-level elements that impact the experience of illness and health care (see Appendix A). We piloted the guide with two AYA cancer survivors from different ethnic backgrounds and gender identities and refined it based on their feedback.

Interviewers employed a patient-led study design [34] that invited participants to direct the conversation by beginning the interview with a quiet reflection prompt that led to the participant’s story (see Appendix A). We used a “grand tour question” that asked participants to talk about their cancer experience broadly [36,37]. As the participants shared their recollections, the interviewer located constructs in the interview guide, noting which construct was mentioned first, and probed for further details. The interviewer exhausted all domains represented in the initial story and then asked questions to elicit descriptions of other aspects of cancer survivorship at various time points (before, during, and after cancer treatment). We documented a range of stories and experiences in these interviews, but clear patterns emerged across all participant narratives despite this variety, demonstrating thematic saturation [38].

### 2.3. Analysis

We applied established criteria for rigor in qualitative analysis to ensure the trustworthiness of our findings [39,40,41]. After each interview, the interviewer created a profile of each participant, summarizing the main themes of their experience across all domains [42,43]. We closely reviewed the profiles of our first 10 respondents and created a comprehensive summary of all themes [43], which we used to inform our preliminary code list. Consistent with the tenets of reflexive thematic analysis, once all interviews were completed, we reviewed our full transcripts in Atlas.ti [44] to ensure that our codelist identified interesting features of the data [19], adding in vivo codes based on new data, refining codes by discussion among our coding team (LS, CD, SR, and MFG), and eliciting feedback from other qualitative experts on our project team (BZ, NC, and MC). We incorporated changes and finalized the code list. The coding team conducted a reliability check on 10% of the transcripts with the finalized code list to establish consistency in approach, designation of units of meaning, and application of codes between coders. Then, the team independently coded the remaining transcripts. A second coder [45] from the team reviewed every transcript to ensure all concepts were captured and to identify possible contributions to the patient-centered model of survivorship care. We used coding memo structures in Atlas.ti to note coding decisions and ideas about the data/conceptual model [2] to inform our results. Our team met weekly to discuss the coding processes and resolve any discrepancies in coding decisions. In these meetings, we discussed our impressions of the data, building rich interpretations [19,20] through collaborative sense-making.

Once coding was complete, we reviewed code density and code co-occurrence at the document level, code family level, and individual code level, paying close attention to themes that were most salient for participants and that would inform a patient-centered model of AYA survivorship care. We wrote detailed code memos analyzing these themes and reviewed the literature to ascertain which elements were not represented by current conceptual models of survivorship care. Those memos inform the results section that follows. The quotations were edited for length and clarity.

## 3. Results

In this qualitative study, we conducted a total of 33 semi-structured interviews. Eight interviews appeared, based on voice and content, to have been completed by the same person. Members of the core study team independently reviewed the transcripts and removed those believed to be fraudulent (see limitations). Ultimately, 25 transcripts were included in the analysis. Most participants were in their 20s and 30s at the time of diagnosis, and the majority (84%) were women. Nearly half of our participants were between 2 and 5 years from treatment, with 20% less than 2 years from treatment and the remaining participants more than 5 years from treatment. The most common cancer types in our sample were leukemia and breast cancer (see Table 1).

### 3.1. Overall Themes

The main themes that emerged from our interviews with AYAs included (1) difficulty navigating the healthcare system and the need for support with care coordination, (2) lack of access to critical ongoing mental health support, (3) need to be connected to AYA peer support and barriers in establishing that connection, (4) importance of empathic and time-sensitive communication throughout treatment, particularly during fertility preservation efforts, (5) support with financial burden, (6) support for quality of life, (7) education about and support with side effects and late effects, and (8) attention to the unique needs of young adults (YAs) as compared to adolescent and adult survivors. These themes arose from responses to interview questions about personal health, including sexual health, personal goals, emotions, social relationships, daily logistics, employment and career, medical care, and planning for the future (see Appendix A, Interview Guide). It is important to note that the themes co-occurred in participant responses. For example, people described difficulties with fertility preservation when responding to questions about provider communication.

### 3.2. Care Coordination and Healthcare System Navigation Support

The need for support with care coordination as participants learned to navigate the healthcare system was a key theme across all interviews. Participants spoke about how difficult it was to advocate effectively for themselves during their diagnosis and treatment and as they made decisions about their care. They reported feeling rushed, inexperienced, and unaware of what questions they could or should be asking of their care teams. Likewise, the importance of clear and empathic patient-provider communication was cited as critical to care coordination.

“So there really wasn’t much time. Or was there? I didn’t know to ask that question. Okay, I know this is growing—is there enough time for me to get a consultation? I don’t know if maybe I could have waited a few days. I just don’t know, because I didn’t know that question to ask... But I just went ahead and signed away because I felt like I was—I hate to say the word bullied, but I felt like I was in a corner. I was like oh my god—this cancer’s bigger than me, just get it out, kill it! Do what you need to do.”—*Participant 1, female, breast cancer, 30–39 years old at diagnosis*.

“I, I mostly blamed myself for my inexperience in hospitals, I guess. But yeah, I felt like people weren’t necessarily completely clear, well, telling me exactly what I had to do. What I should do. Like when I should ask for help or when I didn’t need to, that sort of thing.”—*Participant 2, female, renal cell carcinoma, 30–39 years old at diagnosis*.

Several participants described having to learn to coordinate their own care and how difficult and exhausting it was. One participant spoke of switching from a team with insufficient care coordination to a care team that better met their needs.

“I felt like I had to be the care coordinator. I had to make sure everybody knew what the other was doing. Proactively ask for appointments—like okay, I’m going to have to get radiation next. And they’re like oh, you can wait for that until the week before, and I was like, but what if I don’t like [the provider]? You’re going to put me in a box. So I had to just be proactive to get the kind of care that I wanted to get. And I felt like my care coordinator, which is exhausting.”—*Participant 4, female, breast cancer, 30–39 years old at diagnosis*.

“I was first getting treatment somewhere and I didn’t feel completely taken care of there. As a nurse practitioner, I felt like I was asking—I was supposed to be a patient then, I wasn’t supposed to be a health care provider. So I felt like I was directing my care and I was reminding them of things. It didn’t feel like the right fit for me with my oncologist and the care team, so I ended up after getting a second opinion switching to another hospital.”—*Participant 3, female, Hodgkin’s lymphoma, 20–29 years old at diagnosis*.

Participants in our study spoke about their need for someone on their care team who would treat them holistically and follow up on important aspects of their care. The importance of support with care coordination and healthcare navigation was also highlighted when participants shared how helpful it was when they did have someone on the team to support them in this way. They also emphasized the importance of support tailored specifically to AYAs’ needs and experiences.

“Gosh, that’s really why I became an advocate—I just couldn’t believe the lack of treating me as a holistic person. I understand that I guess to be an oncologist you’re going to meet patients who ultimately die from it, and I get that they’re trying to make sure that you don’t die, and that is of course great, you kind of need that. But what about a nurse navigator or even like the nurse? There was no follow up... there needs to be a middle person. Whether it be that nurse or that social worker, and it should be mandatory that every AYA... have an initial conversation [with them] and then determine if you want to work with them...The follow ups just go through the cracks.”—*Participant 1, female, breast cancer, 30–39 years old at diagnosis*.

“I felt like my oncologist was very good at giving me medications to deal with nausea and other side effects when I needed them...But I had to research online what are things that I could use and then go and ask for it, as opposed to someone presenting me with “these are all the resources” or “these are things you should consider, let us know what you need”. I felt like the latter would have been much more helpful. I went to [other specialty cancer centers, and] both of those hospitals did provide that. Like “here’s your coordinator, here’s a whole pamphlet, here’s all the resources we have. Here’s how you use each one”. So I thought that was really cool.”—*Participant 4, female, breast cancer, 30–39 years old at diagnosis*.

### 3.3. Mental Health Support

AYA survivors in our study described the serious, unmet, and ongoing need for mental health support across the cancer care continuum. Every participant described dealing with mental health challenges, including anxiety and depression, as a result of their diagnosis and treatment. Many also described a loss of control and an increased sense of isolation.

“Definitely anxiety, depression for sure. I think those would be the biggest two that I’ve had to deal with. It’s an everyday struggle … Anxious about my cancer getting worse or also having cancer in my family or friends, since I already know what it feels like, having cancer. I wouldn’t want any of my loved ones to go through the same thing.”—*Participant 6, female, breast cancer, 30–39 years old at diagnosis*.

Participants shared that they need additional mental health support and resources both during treatment and now after treatment is over; many described cancer treatment itself as traumatizing.

“Cancer is trauma, and even though a lot may not equate it with that term, because they just don’t know, a lot of us have PTSD. And that’s not talked about enough… every experience in the AYA community matters. So that might be why someone would not [talk to a researcher about their cancer experience], because they might feel like you could talk to someone better. It’s really about insecurity, but also too how they’ve been treated throughout their treatment. It can be hard to discuss and be traumatic. I can now verbally talk about it without bursting into tears, but not everyone can.”—*Participant 1, female, breast cancer, 30–39 years old at diagnosis*.

“Obviously having cancer kind of like fucks you up mentally. But I’ve been going to therapy, I actually take an antianxiety med now.”—*Participant 8, female, Hodgkin’s lymphoma, 15–19 years old at diagnosis*.

A common source of solace in the face of mental health struggles was a connection with other AYA cancer patients and survivors. Peer support (see section below) was instrumental in supporting mental health and reducing feelings of isolation. Therefore, connecting newly diagnosed AYAs with existing AYA support groups and networks may have a significant impact on reducing their mental health burden.

“Like I thought, I thought I was alone for like five years … Post treatment I actually had a really bad depressive episode, because I was just in such despair because I thought I was alone and no one else was like me. And I did hours of searching and finally found a couple of organizations that led me to other things. But I would have liked to have those resources [earlier], I wouldn’t have felt so alone.”—*Participant 8, female, Hodgkin’s lymphoma, 15–19 years old at diagnosis*.

“I actually learned about the support groups from Instagram … just as a young Black woman, [it was important] to see other women of color that were young and that looked like me, because I was not seeing that at my cancer center. So that was a huge support for me. Also, just by sharing my story, it allowed me to pay it forward to other young adults and also inspired me to get involved in advocacy work.”—*Participant 9, female, breast cancer, 30–39 years old at diagnosis*.

### 3.4. Peer Support and Making “Cancer Friends”

AYAs experience intense feelings of isolation from often being the only young patient experiencing cancer, as they interacted with few other AYA patients during their treatment, compounding their difficulties.

“It’s bad enough I’m an AYA, it’s bad enough I’m Black, it’s bad enough I’m a woman, it’s bad enough that I am an only child. I feel like all of these things were hitting me—and I have cancer, and now I literally have no one? It’s been hard.”—*Participant 1, female, breast cancer, 30–39 years old at diagnosis*.

“So, I think at the time the quintessential experience of being the youngest person at the cancer center in the waiting room, you know, not seeing anybody else my age unless they were in a caregiver capacity... And just feeling like I was the only person my age that had cancer and was getting treatment. And so the experience was very different when you are under 40. I didn’t know other people that had gone through that at the time.”—*Participant 10, male, testicular cancer, 30–39 years old at diagnosis*.

Nearly every participant mentioned the importance of support from fellow AYA survivors, which can reduce or mitigate feelings of isolation and help alleviate concomitant depression and anxiety. Peer support was also essential for managing symptoms. Other AYA survivors with similar cancers or similar treatment plans can provide valuable expertise about how to manage side effects and commiserate with experiences that are unusual for young adults who have not experienced cancer. Also, symptom management was not always provided compassionately by providers, which compounded the need for peer support.

“As I was nearing the end of chemotherapy, I was feeling like I couldn’t really talk to my friends the same, and I didn’t really have people to relate to, and I felt like an astronaut. My brain was foggy, I really wanted to talk to someone about [my side effects and stuff] without worrying people. I remember Stupid Cancer was the big [AYA organization] at the time, and I saw that they had in-person Meetups. I decided to go … and then I instantly was like oh, maybe this [is] a window into a community I didn’t even know existed. I didn’t picture people in their 20s and 30s with cancer hanging out before this. That was the beginning of making cancer friends, [we have fun but] also if someone does need to vent about their situation, treatment, insurance, or relationships going away because of cancer, you’re the perfect [person] to talk to about it.”—*Participant 11, male, testicular cancer, 30–39 years old at diagnosis*.

“I went through a lot of side effects. I literally had the motherlode of side effects and what was very hurtful was when my oncologist would be like yeah, you know, a lot of patients get that. Well, it’s my first time seeing my tongue turn black, so you might want to have some sort of—I don’t know, like compassion for how freaked out I would be. Even my throat would swell and I had difficulty swallowing. ‘Oh, I’ve seen it before, I’ve seen worse.’ Well, I’ve never seen worse.”—*Participant 1, female, breast cancer, 30–39 years old at diagnosis*.

Accordingly, AYA survivors would have appreciated specific directions to appropriate peer support by their care team (in-person peer support groups, online peer support groups, or online cancer support organizations). Many participants who eventually found their own way to these groups reported that knowing about them sooner and having help determining which would be most suitable would have eased the mental health and symptom burdens inherent in their cancer experience. Several also described feeling out of place at their hospital’s general cancer support group, as it was populated predominately by older adults, who were facing a very different set of concerns.

“I wish that that there was an AYA program at the hospital to tell me about these resources. To tell me like, hey, there’s a Gilda’s Club, it’s 10 to 15 min from here. There’s a meeting once a month. You can go and meet people your own age. It’s safe. People are really cool. Check it out. And now you can join these virtually. Just having somebody to say to me that is totally normal to feel that way. There are other people your age that get treatment here and you can meet them. That would have been really awesome.”—*Participant 10, male, testicular cancer, 30–39 years old at diagnosis*.

“I think just introducing for patients, that adolescent young adult oncology exists, and there is support out there for AYA’s. I didn’t really dive into the AYA support community until after treatment and got connected to a lot of resources and a lot of friends that way. But I think if I had known that resources like that existed while I was going through treatment, it would have been helpful just to know that I wasn’t alone and all these amazing organizations exist.”—*Participant 12, female, osteosarcoma, 15–19 years old at diagnosis*.

### 3.5. Empathic Communication about Fertility Preservation

Participants shared that communication around fertility preservation was presented too close to diagnosis, without context as to why preservation was important, in a rushed manner, or not presented at all. Although participants recognized that these were time-sensitive conversations that often could not be deferred, processing so much information at once was overwhelming. It was difficult for many AYAs to balance the immediate urgency of cancer treatment with the need to seriously consider fertility treatment; these challenges were particularly acute for patients assigned female sex at birth, for whom the logistics and timing of fertility preservation are more complex. Some patients faced significant barriers to preserving fertility, such as cost and time, which varied depending on cancer type, stage, and treatment plan. AYA survivors described the importance of different members of their care team speaking empathically with them about this.

“When I got diagnosed in the hospital … they had brought in a blood specialist and he described leukemia to me … After he left one of the interns immediately asked me, like so do you have any kids? And I was like no. And he was like, have you thought about freezing your eggs? And I’m like, dude, this dude just told me about cancer, like I haven’t, I can’t talk about kids right now like. You know?”—*Participant 13, female, leukemia, 20–29 years old at diagnosis*.

“The timing was rushed because it was overwhelming. I feel like if you sit down with anybody, man, woman, whatever, and tell them you might not be able to have kids, that’s pretty heavy and something you want to sit with. And … it’s not like it was free to go get the sperm banking done and have it stored. But I was like well, if I don’t do this, that might be it, I might never have kids. Even if I don’t want them at the moment, taking the option off just seemed scary. So yeah, I would have liked to have had more time.”—*Participant 11, male, testicular cancer, 30–39 years old at diagnosis*.

Several patients described the rushed nature of discussions about fertility preservation and the lack of information they received. They indicated that, even in situations in which the nature of the cancer and its treatment do not allow time to attempt fertility preservation, empathic and repeated conversations about options, resources or supports and counseling to allow patients to process or grieve are critical to good care.

“Everything for me happened within like three days, so there was no, no ability to like, I don’t even know what it’s called. But to … freeze my eggs, I didn’t have that option because of the type of cancer I had everything had to be done so quickly. The only thing I was told in regards to fertility is you may not be able to have kids. There’s a high likelihood with the chemotherapy you are receiving that you may not be able to have children after this. There was no offering of like any type of resources. I only found that out afterwards, [about] all like the different type of programs for patients.”—*Participant 15, female, leukemia, 20–29 years old at diagnosis*.

“We talked about [fertility preservation] in [my support] group before and I guess, well, I mean for guys it’s easy, so they’re super on top of it as far as when we spoke about it. But a lot of [women] who were in similar positions to me where it was all just really sad. From my experience [the doctors] were like, okay, you’re here now, here’s your doctor, here’s your treatment. Oh, by the way there’s this [fertility preservation option], we kind of want to get started right now, so could you just not [have kids] … It wasn’t a huge deal, but I was a little sad.”—*Participant 14, female, leukemia, 20–29 years old at diagnosis*.

“There should have been a follow up call [after my diagnosis]. Because that was a really intense moment. My first time as the patient … Why wasn’t there a follow up? Like hey, I know you just heard a lot of information, let’s talk about this. I feel like I should have at least been required to get a consultation with an infertility specialist, even though it wouldn’t have been covered under my insurance. I feel that conversation should at least have been had so they could make sure I was really making the best decision for myself at that time. Sorry, I get really passionate and very angered about it.”—*Participant 1, female, breast cancer, 30–39 years old at diagnosis*.

Several participants reported communication failures or misunderstandings with their healthcare providers. This lack of communication led to distress regarding unexpected side effects, such as fertility loss. One participant discussed how they had to seek out resources and information about their cancer treatment and potential side effects because it was not proactively communicated to them by their providers. Several participants reported such significant failures of communication that the side effects they experienced were unexpected.

“I lost my fertility. No one prepared me for that. I didn’t receive initial counseling going into that surgery or coming out of it. I didn’t expect to experience that kind of grief, because I was single all this time, and childless, and now I am chronically single and barren forever. None of my doctors cared to see how that would affect me.”—*Participant 1, female, breast cancer, 30–39 years old at diagnosis*.

“I don’t really have trouble communicating with [doctors]. I’m a lawyer and I did a lot of research, so I generally got the comments that ‘oh, you’re so knowledgeable, you’re an easy patient.’ [But] I don’t think they necessarily answered all my questions, or gave me all the resources that were available, or were upfront about side effects, which I found frustrating…[the doctors failed] to mention fertility resources [so] I found my own stuff … I certainly wouldn’t say I got most of my information from my oncologist, but I found it in other places.”—*Participant 4, female, breast cancer, 30–39 years old at diagnosis*.

However, a few described having good communication with their physicians as they navigated these difficult decisions. Providing the necessary context and weighing the complex natures of different patients’ needs were both crucial to these successful patient-provider conversations. Other care providers (e.g., social workers, nurses) played an important role in discussing fertility preservation options or finding funding for the fertility preservation process.

“My oncologist is very respectful of my wishes in terms of wanting to have another baby … but then [she] also wasn’t afraid to tell me, you know, we can only do one round of harvesting your eggs, because it’s not safe to do more. She did a really good job acknowledging my dream and weighing that accordingly, [so] I’m not risking life … but I’m still able to try to, you know, preserve my fertility before having this definitive surgery.”—*Participant 5, female, ovarian cancer, 30–39 years old at diagnosis*.

“Before I started chemo, my social worker came to talk to me in the hospital room and she just wanted me to know like hey, your doctors want you to do chemo, but you don’t have to do it right now, you can work on the fertility thing, if it’s important to you. So she made me feel comfortable that it was okay to delay the treatment.”—*Participant 7, female, leukemia, 20–29 years old at diagnosis*.

### 3.6. Financial Burden and Need for Support

Participants described the need for financial support along every step of their cancer journey, from selecting an insurance plan and treatment center to transparency about and help to manage the cost of treatment and other essential services. Almost all AYA survivors we spoke with discussed the financial burden of treatment, alongside a substantial burden caused by the labor of navigating the health system itself.

“We needed help, we had help from family and friends, but again, the financial burden … is just a nightmare. You got the financial burden, you got the paperwork. You’re supposed to be focusing on your health.”—*Participant 5, female, ovarian cancer, 30–39 years old at diagnosis*.

“I worked in fine dining and didn’t have any insurance … And then the diagnosis alone racked up I think tens of thousands of [dollars in] debt and I was just through biopsies and scans and you know. I was going to, which is laughable, but it was called free clinic. It took a long time before I was diagnosed; go get bloodwork, come back in two weeks, schedule another appointment for two weeks later. And debt was mounting.”—*Participant 16, male, Hodgkin’s lymphoma, 20–29 years old at diagnosis*.

“I probably know more about the American health services than I ever wanted to know … it’s just not the way I would have liked to have learned it.”—*Participant 8, female, Hodgkin’s lymphoma, 15–19 years old at diagnosis*.

Many AYAs younger than 26 years old with whom we spoke received insurance through their parents. As they approach their 26th birthday, the burden of finding their own insurance and navigating health care payments increases, which is an additional burden that would benefit from specific support.

“With my age I am able to be on my dad’s insurance and it is a really good insurance plan. So it hasn’t been like insanely expensive or anything … But as I approach my 26th birthday, the cutoff [of staying on my parents’ insurance], I have lots of concerns with finding good health care on my own.”—*Participant 14, female, leukemia, 20–29 years old at diagnosis*.

### 3.7. Quality of Life

Most of our participants discussed the disruption to life goals, employment, and living situations they experienced as they received their cancer diagnosis and treatment. Disruption of life goals included disruptions in achieving professional goals, such as a license or special certification.

“When I was first diagnosed I was studying for a board license for civil engineering. I was still thinking I’m going to be in chemo for eight hours, I’ll have a lot of time to study at the hospital. It wasn’t like that at all. That’s when I was in denial, and I think after that, that’s when depression hit me. I was like you know what? It’s over, I’m just going to keep my job now. There’s no way I can study for the exam … Sometimes in my back of my mind I’m still thinking I want to be a licensed engineer and all I have to do is pass that exam. I start dreaming that when I pass the exam, I’m going to get my promotion and travel more, which I used to do before diagnosis … I guess career-wise I still think about getting my license, even if I don’t keep working in the engineering field, I want to feel accomplished. I want to be able to say even through or despite cancer, I was still able to accomplish that.”—*Participant 6, female, breast cancer, 30–39 years old at diagnosis*.

“So because I got sick, at least with my internship hours, I could have been done last December. But I was going through treatment. And my friend and I were collecting hours and going to school at the same time. She already finished herself, got certified, she’s my boss right now. She’s my supervisor. We were like at the same level, she’s already above me. So and she doesn’t treat me any lower, but I’m still a little upset sometimes because I could have been there by now if I hadn’t gotten sick.”—*Participant 13, female, leukemia, 20–29 years old at diagnosis*.

Some participants shared how cancer and survivorship care disrupted and continues to disrupt their employment. Some participants felt forced to quit their jobs, while others struggled to find ways to take time off for cancer-related and other medical appointments.

“I’ve been a dog groomer on and off for about 10 years. And I when I was finally able to get back into work [right after my surgery], I felt like they didn’t understand what I was going through. Like I was very anxious, and there’s a lot of sounds in a grooming salon. And it was really putting me on edge. And I started to wear earplugs to deal with that. And then I started getting like looks from my coworkers and like I just started to feel less and less welcome there. And I just gave up on it and I ended up quitting that job. I just didn’t feel very good there anymore.”—*Participant 2, female, renal cell carcinoma, 30–39 years old at diagnosis*.

“I did officially go back up to my regular hours, but there are some days that I take time off for appointments. I try to schedule for example my scans in one day, for example, so I only have to take one day off whenever I can…It’s not just cancer that we deal with, we still have to deal with what other people go through as well, for example taking time out for dental and eye doctor appointments. I still have to take time off for that.”—*Participant 6, female, breast cancer, 30–39 years old at diagnosis*.

Many participants discussed disruptions to living situations and families caused by needing to move to new locations to pursue treatment.

“I had never been to the hospital before. And so I had to go through getting my diagnosis. Going through all these different procedures. And every one alone. They transferred me because they didn’t have the resources where I live to treat me. They transferred me to Houston, so my life got uprooted. My job put on hold. I had to move about five hours away so I could get treatment.”—*Participant 13, female, leukemia, 20–29 years old at diagnosis*.

### 3.8. Information about and Support Mitigating Side Effects and Late Effects

Over two-thirds of the people we spoke with discussed their experience with side effects and late effects. They were concerned with receiving treatment with side effects that are tolerable across their lifespan, effects related to serious conditions such as osteoporosis and nerve damage, and permanent disability. They noted that these concerns are more pressing for AYAs than traditional oncology patients.

“The important elements for young adult cancer care compared to the typical cancer patient that you think of, like 50, 60, 70, they’re worried more about the here and now, and they don’t necessarily have to worry about side effects 20, 30 years down the road, because life expectancy, they won’t be there. I was diagnosed at 25. God willing, I’ll be alive for 50 more years beyond that. I don’t want to be dealing with side effects for years on end, so if there’s an option that’s a little bit more conservative treatment, which will possibly result in less side effects but maybe instead of saying it’s 100% certain, it’s 80% certain. That’s a 20% difference, so I think addressing that in terms that are easily understood by young adults, and also not in a talk down to manner, is super important.”—*Participant 17, male, testicular cancer, 20–29 years old at diagnosis*.

Participants discussed their concerns about experiencing side effects or late effects that impacted their dental health, joint and bone health, skin issues, chronic pain, and permanent disability. Many expressed that these effects were surprising, that they wished they had known what to expect, and that they felt unprepared to cope with them.

“Oh, and then the thing I always forget are the other secondary effects of treatment. I had to have both shoulders and both hips replaced, and I had no idea that was going to be in my future whatsoever, at the time of treatment.”—*Participant 18, female, leukemia, 20–29 years old at diagnosis*.

“I have osteoporosis and I’m not even 25 yet, so that’s kind of concerning for the future.”—*Participant 14, female, leukemia, 20–29 years old at diagnosis*.

“The one thing I do deal with is, because of all the surgery I’ve had, I have chronic nerve pain, nerve damage, so that’s not fun to deal with. I wish I would have known that it was a possibility, because I was not told that it was a possibility that this could happen.”—*Participant 19, female, sarcoma, 15–19 years old at diagnosis*.

The participants who experienced permanent disability spoke of their struggles, sometimes over the years, to get adequate care for the damage done to their bodies by treatment. For some, the infrequency of specialist support is a challenge with medication management and understanding the condition of their bodies.

“I’ve got major issues with the majority of my organs. I have liver damage. I have heart failure. I was in a wheelchair for a while. I was on bedrest for a very long time right after everything. I am disabled. I am on disability. And I do not have the energy I once did. Napping and every couple days just being totally exhausted is kind of part of my life.”—*Participant 20, female, leukemia, 30–39 years old at diagnosis*.

“I have permanent damage—I don’t feel my feet, my toes from the upper balls to my toes. Sometimes the numbness goes up my legs… and I’ve fallen, actually almost fractured my ankle in January because I didn’t feel my foot. It was so sudden and severe, and … no one seemed to take it as seriously as I did, which is frustrating.”—*Participant 1, female, breast cancer, 30–39 years old at diagnosis*.

### 3.9. Attention to the Unique Needs of Young Adults

While we did not specifically ask about the needs of young adults with cancer, this theme arose in most of our interviews. We heard that young adults feel isolated as young people with cancer, both in treatment and socially.

Several people mentioned benefiting from or wanting specialty care tailored to the unique needs of young adults, which to them meant having providers who were knowledgeable about their needs, receiving age-appropriate information, and having support for the life changes experienced by young adults.

“[My center had] an AYA program. Granted, they have so much volume because they have a special unit, so I think volume begets resources. But they have providers who are knowledgeable and not just oncologists, but lots of different providers who are knowledgeable about issues that AYA’s face, especially fertility. Sometimes we respond differently to drugs. If every center could have somebody who has a special research focus, to keep up to date on AYA’s. Or a pamphlet, a website, that even would have been helpful. I feel like there’s many ways to skin the cat, but it’s just providing age-appropriate information.”—*Participant 4, female, breast cancer, 30–39 years old at diagnosis*.

“But I definitely wanted more [young adult] support specifically. And not just in general cancer support, I went through this huge ordeal; it’s completely life changing. And I just, to me the more support I’m getting I feel more in control and I have more power.”—*Participant 5, female, ovarian cancer, 30–39 years old at diagnosis*.

## 4. Discussion

Our findings regarding the lived experiences of AYA cancer survivors confirm the results of previous studies [46,47] and extend current conceptual models of AYA survivorship care [1,3,4,5,6,7,8,9,10,11,12,13,14,15,16,17,18]. Many of the existing models focus on discrete clinical AYA survivorship needs, but we offer a conceptual model of survivorship care (Figure 2) that addresses the holistic needs of AYA survivors. As in other studies, we found that receiving a cancer diagnosis as a young adult is a life-changing event with long-term implications and that the timing and delivery of healthcare information and support need to be improved [9,11,17]. We identified a pronounced need for mental health and peer support; these elements were our densest codes, discussed by each of our participants. We build upon earlier models by, therefore, framing mental health and peer support as central to all aspects of cancer care and as needs that merit repeated support over time.

Of greatest note, our patient-centered model reframes cancer and survivorship care as cyclical rather than linear, recognizing that, for our participants, the experience is not confined to a treatment schedule and affects many aspects of their mental and physical well-being for years following diagnosis. Furthermore, we found gaps in current models of care that might be addressed by improved care coordination, a call that is echoed in the literature [48,49,50]. Our interviews indicate that AYA cancer survivors would benefit from continuing supportive navigation that prioritizes care coordination, mental health support, peer support, fertility preservation, financial burden, quality of life, side effects and late effects, and attention to the unique needs of young adults.

Participants in our study shared that the experience of cancer treatment for AYAs, even at well-resourced cancer centers, is often isolating, overwhelming, and traumatizing. While oncology practices and these conceptual models, in turn, tend to focus on the health of the body and portray “survival” as an endpoint, our participants emphasized the inadequacy of this framing. Many participants insisted on the critical need to be attended to as people and not only as patients throughout treatment and their lifetime and to have survivorship needs treated with as much care and coordination as their cancer treatment.

Our results highlight the importance of ongoing and repeated efforts to address patient needs at multiple time points during the cancer care continuum. Studies have shown that investing in care coordination can reduce care costs and improve efficiency [51] and that AYAs appreciate AYA-specific support and care coordination [48,52,53]. Our participants identified that many of the issues they face are structural in nature. Acknowledging that healthcare systems, especially in the United States, are often monolithic and provide care in ways that disproportionately isolate and often harm vulnerable populations [54,55], including people who identify as sex or gender minorities (SGM) [56,57,58], BIPOC [56,59], or who experience pre-existing mental health conditions [60,61], can compel us to provide the type of support that will center patients as whole persons and meet their needs [62].

The model of AYA survivorship care that we propose evaluating and testing (Figure 2) locates the patient at the center, surrounded by their social support networks (family, friends, community), which may vary in strength and stability over time and across patients. AYA-sensitive navigation support could connect patients to the various resources they need and help them navigate the health systems they encounter. The care elements surrounding the patient represent the broad areas of the health system that the patient will interface with and a full picture of where support is needed. Although the icons representing each care element are separate, our model emphasizes the interconnection between each of these elements. As our participants explained, having financial support and peer support would ameliorate mental health burdens, for instance, and ongoing clinical care ought to include multiple conversations about fertility concerns, even when the timeline necessary for such intervention is short. We note that empathic, timely, and ongoing provider communication is key in each of these elements. We offer a model that includes the components of patient-centered care identified in this study’s interviews and the broader literature and that emphasizes the continued need for this care beyond the final treatment.

Each of the elements is described in detail below.

### 4.1. Care Coordination and Healthcare System Navigation

Care coordination is an overarching theme in our model. Robust support with care coordination and healthcare navigation is necessary to facilitate patient’s access to needed resources and providers, including mental health care and peer support [63,64,65]. The findings from our study suggest that healthcare systems should consider how to structure care to support the unmet needs of many AYA survivors, whether in person or virtually, across healthcare contexts. A possible implementation of this model could include an AYA support specialist, similar to a Child Life Specialist seen in pediatric cancer settings or the AYA Life Specialist seen in many AYA-specific clinical settings [48,66,67]. This specialist would be aware of the particular needs of the individual patient they are working with, checking in regularly with the AYA patient to ensure their understanding of their diagnosis and treatment options, connect them to needed resources, providers, and social support, and normalize the need for and help facilitate access to mental health support.

### 4.2. Mental Health Support

The need for mental health support was a major theme in our interviews; mental health is negatively impacted by cancer care in general [68,69] and exacerbated by being a young adult living with an illness most commonly associated with older adults [70,71,72]. Our interviewees emphasized the need for mental health support throughout the cancer experience—at the time of diagnosis, during treatment, and after treatment. Our data show that regular and repeated time points for the offering of mental health support need to be prioritized in patient-centered AYA care. Additionally, the lack of support with healthcare navigation causes serious detriments to mental health for AYAs experiencing cancer [69,73].

### 4.3. AYA Peer Support

Impacting AYA mental health is the connection, or lack thereof, to peer support from other AYAs who have experienced cancer. Challenges from cancer and cancer treatment add to a long list of additional complications AYA cancer survivors are managing given their life stage (i.e., time off from work or loss of a job, parenting or relationship management, financial planning, etc.), and contribute to feelings of isolation from peers without cancer, who may have little frame of reference for the magnitude of the decisions young adults with cancer face and the amount of labor involved in being a cancer survivor. There is ample literature on the extreme isolation AYA cancer survivors experience [14,74,75]. There is also increasing attention being paid to the importance of social support from peers for help with symptom management [76,77], commiseration and validation [78], and other quality-of-life issues [76,79], issues our participants mentioned repeatedly. One study found that AYA survivors ranked opportunities to connect with peers with cancer as more important than support from family and friends [78]. Since many AYAs are not aware of in-person or virtual AYA peer support networks during treatment, support is needed to facilitate such connections [63,64,65].

### 4.4. Empathic Communication about Fertility Preservation

The importance of oncofertility support for AYAs is crucial [3,4], both during fertility preservation and family building [80,81]. Empathic, timely, and repeated conversations with providers are especially important [76]. Improved care coordination could help structure conversations between the care team and the patient about fertility and other treatment options to minimize confusion, overwhelm, and decisional regret, alongside connecting the patient to fertility preservation and/or family-building resources if appropriate.

### 4.5. Financial Burden

We heard that the financial burden of cancer care is especially difficult for AYAs, who face the challenge of paying for treatment and care at the time they are receiving an education or just beginning their careers and are not yet financially stable [6,72,82,83]. Though the Affordable Care Act (ACA) expanded dependent coverage on a parent’s health plan until a young adult’s 26th birthday, many AYAs face the added burden of losing insurance coverage when they turn 26, and this may be especially problematic for those living in Medicaid non-expansion states [84,85,86]. As a result, many AYAs forgo insurance, with adults ages 26–30 being the most uninsured age group in the US [87]. Because they often cannot afford premium health insurance, potentially lack access to employer-sponsored insurance coverage, or elect to a less comprehensive, low-cost health insurance option as they had the expectation of being healthy, this, in turn, leads to the increased financial burden of treatment, and care [85,86]. AYAs need support as they transition off of their parent’s insurance and may benefit from connections with exchange navigators/access assisters who help people enroll in marketplace plans and Medicaid.

### 4.6. Quality of Life

The data presented above describe specific care elements important to AYAs. There may be other components that foster quality of life particular to the patient, such as their available resources, their treatment plan, and their illness trajectory [6,8,14,18,73,88]. For example, AYAs may benefit from support with education and career planning, navigating relationships with family and friends, negotiating time off from work, and help with childcare, among other needs [73,88]. We include quality of life in our model to indicate that patient-centered care would support the important elements that comprise quality of life for each patient.

### 4.7. Education and Support Regarding Side Effects and Late Effects

Another element that occurred across themes was the importance of receiving ongoing education and information about side effects and late effects. The importance of support managing side effects was mentioned when participants spoke of the importance of adequate care coordination, peer support, and empathic communication about possible side effects and late effects. About 40% of AYAs experience long-term side effects, though cancer-related disability is not yet part of conversations about clinical guidelines and cancer research [89]. Patient-centered AYA survivorship care would include continued support, that is, multiple care touchpoints over time, for side effects and symptom management [90].

### 4.8. Attention to the Unique Needs of Young Adults

Though we have discussed the care needs of AYAs as a cohesive group, it is important to note that young adults may have specific and unique needs within this population [73,91]. Our study included participants who were diagnosed with cancer as adolescents; adolescents are often incorporated into pediatric care units, which usually have quality-of-life and care coordination specialists [67,92,93]. Young adults, by contrast, are most often treated alongside older adults, but they have several specific, unique care needs and experience critical gaps in care delivery [94]. The differences in the need for age-specific resources and support for young adults is an area that would benefit from further research [67].

### 4.9. Limitations

An important limitation of this study is that, despite our efforts to enroll a diverse pool of participants, our interview sample was primarily white and female. Achieving success in the inclusion of racial and ethnic minority groups [95] and those who may identify as a sex or gender minority (SGM) among the study participants will require planning and budgeting for the implementation of appropriate recruitment and retention. Costs for implementing evidence-based strategies for achieving more diverse samples must be figured into study budgets. More research to determine how to successfully recruit a more diverse sample of AYA cancer survivors is needed.

Most of the people we interviewed were on the older end of the AYA age range and had been diagnosed during young adulthood rather than adolescence; however, this is consistent with the burden of cancer incidence increasing across the AYA age range. Our interview guide did not include questions about where patients received care or the specific makeup of their care teams, so while these details often came up in the course of the conversation, we were unable to distinguish systematically between treatment and care received at, for example, an NCI-designated Cancer Center versus a community hospital. We also note that we recruited through various online support groups, so our sample may overrepresent AYA cancer survivors who sought out social support and might have different perspectives or experiences than cancer survivors who did not proactively seek out social support.

Finally, we had several cases of identity misrepresentation in our sample. Based on transcriptionist and interviewer input, we became aware that several of the interviews were likely completed by the same person. Three study team members (MF, CD, and MFG) independently reviewed audio files and unanimously identified the repeat interviewees (n = 8). Another team member (JC) then reviewed all audio files (blinded to suspected repeat status) and also identified the same repeat interviewees. Those audio files, transcripts, and interview notes were excluded from the study. There is a need for studies recruiting via agency or online sources to address the possibility of falsely identified participants [96,97]. Studies that broadly advertise rather than doing targeted recruitment are sometimes able to reach a more nationally representative sample, though the risk for identity misrepresentation increases, as we experienced in this study.

### 4.10. Implications for Cancer Survivors

To meet the care needs discussed above, care teams could consider the inclusion of an AYA specialist or multidisciplinary AYA team who would be able to support care navigation and facilitate the closing of some of the care gaps highlighted above. AYA specialty care is already being offered as a standard of care in other contexts and has been shown to positively impact patient experiences of communication and support [48,66,67]. It may be infeasible for an already overburdened system to add all these critical aspects to the workflow of physicians and nurses. Instead, care teams could consider the role of an AYA support specialist or care navigator as a care team member to provide patient-centered AYA cancer care. This type of allied care professional could also provide virtual support and would be especially useful in resource-limited settings or oncology care clinics without the types of care team members or resources important to support AYAs. Such an intervention would warrant testing for both feasibility and effectiveness to see if that role can help to improve AYA care experiences within the specific areas listed above and support better health outcomes for this vulnerable population.

## 5. Conclusions

Our findings suggest that a patient-centered approach to AYA survivorship care could incorporate the identified care needs specific to AYAs. To help determine and address these needs, AYAs may benefit from continuous supportive navigation across each of these domains, repeated over time.

## Figures and Tables

**Figure 1 cancers-16-03073-f001:**
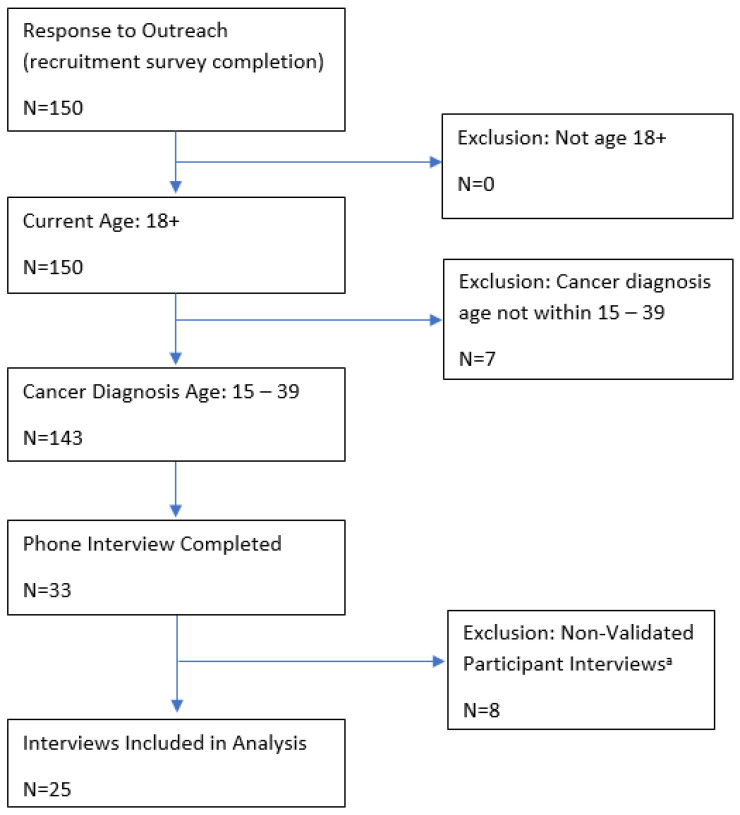
Inclusion criteria for study participants. ^a^ See the Methods section for a description of how the non-validated participant interviews were identified.

**Figure 2 cancers-16-03073-f002:**
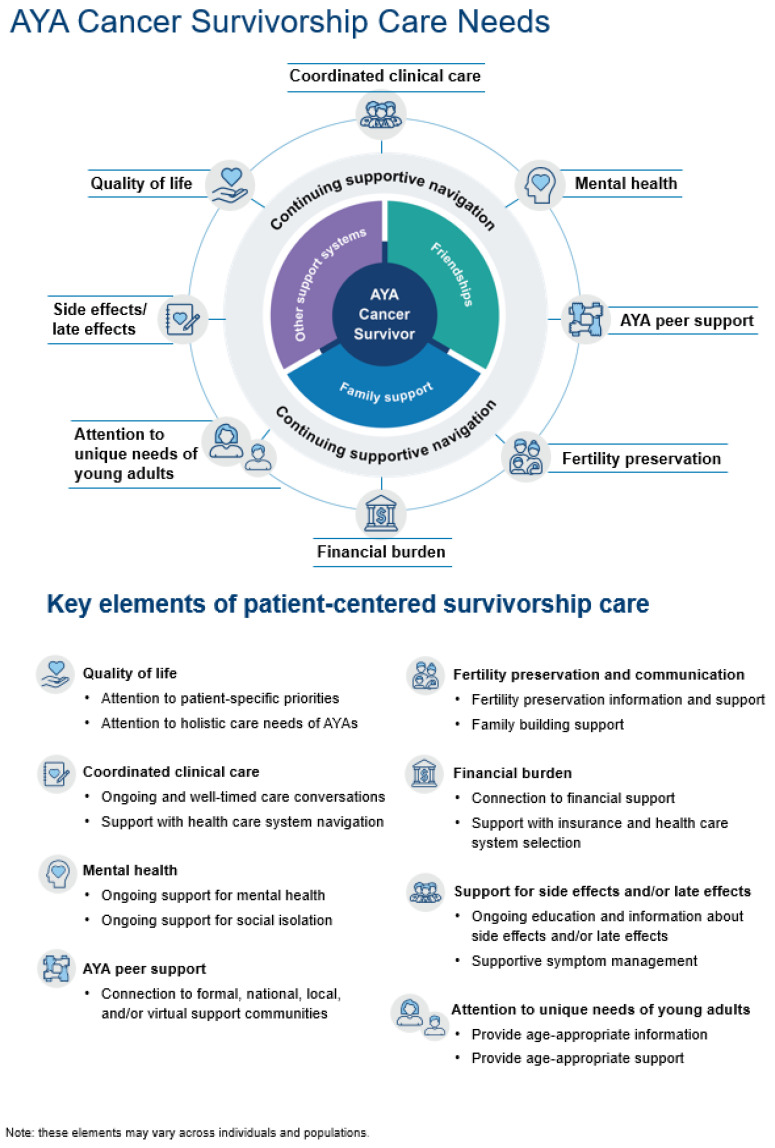
Patient-centered AYA survivorship care conceptual model key elements.

**Table 1 cancers-16-03073-t001:** Demographic characteristics of interview participants (N = 25).

	Number (%)
**Sex assigned at birth ^a^**	
Female	21 (84)
Male	4 (16)
**Race**	
White	19 (76)
Black	2 (8)
Middle Eastern/North African	1 (4)
Other ^b^	3 (12)
**Ethnicity**	
Hispanic/Latinx	6 (24)
Not Hispanic/Latine/x	19 (76)
**Current Age (years)**	
20–29	8 (32)
30–39	12 (48)
40–49	5 (20)
**Age at First Diagnosis (years)**	
15–19	4 (16)
20–29	10 (40)
30–39	11 (44)
**Years Since First Diagnosis**	
Less than 2 years	3 (12)
At least 2, but less than 5 years	8 (32)
At least 5, but less than 10 years	11 (44)
10 or more years	3 (12)
**Years Since Treatment**	
Less than 2 years	5 (20)
More than 2, but less than 5 years	12 (48)
More than 5, but less than 10 years	5 (20)
10 or more years	3 (12)
**Cancer Type**	
Breast	5 (20)
Chromophobe Renal Cell Carcinoma	1 (4)
Hodgkin’s Lymphoma	4 (16)
Leukemia	7 (28)
Lung	1 (4)
Myelodysplastic Syndromes (MDS)	1 (4)
Osteosarcoma	1 (4)
Ovarian	1 (4)
Sarcoma	1 (4)
Testicular	3 (12)

^a^ We did not explicitly ask about gender. All people who responded to a question requesting preferred pronouns provided pronouns that corresponded to their sex assigned at birth; 2 people did not answer. ^b^ Other: 1 participant indicated Mayan, 1 participant indicated Latine/x only (did not indicate race), and 1 participant indicated they identified as Black, white and Latinx.

## Data Availability

The participants of this study did not give written consent for their data to be shared publicly. Therefore, the data generated and analyzed during this study are not publicly available. We include our interview protocol in the Appendix A.

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
