# Peer review of "A Patient-Centered Conceptual Model of AYA Cancer Survivorship Care Informed by a Qualitative Interview Study"

_cancers, 2024, doi:10.3390/cancers16173073_

Round 1

Reviewer 1 Report

Comments and Suggestions for Authors

This manuscript presents a conceptual model of cancer survivorship care. It captures elements of the influence of symptoms and interventions, as well as what is important to the individual as part of their social & cultural groups.

I have the following queries and suggestions for the author's consideration:

- include research design in the title.

- include mean (SD), range of ages of participants at time of diagnosis in the abstract.

- young adults is typically up to 25 years of age. Please explain how up to 39 years is considered young adult.

- please justify rationale for N=30 interviews sought.

- having a codebook that is applied across interviews is not consistent with reflexive thematic analysis approach as described by Braun & Clarke, but more so with qualitative content analysis potentially. Please check.

- Whilst the quotes are really interesting, there are too many making the manuscript very long to read. Also, quotes need to be in italics and indented, so that they are distinguishable from the manuscript text.

- Please only use a quote once. For example, Participant 10.

- what is not clear to me between the text and the visual representation of the conceptual model, is what was identified as proximal elements on AYA cancer survivorship and distal elements? From the text it appeared the proximal elements were mental health and peer support, but this isn't clear in the model. Also, what elements change over time (or do they all?) and what is the impact of the environment and the individual (i.e. typically included elements in a conceptual model).

Reviewer 2 Report

Comments and Suggestions for Authors

The authors addressed a very interesting and innovative topic. The manuscript is well written and coherent in its different parts. I appreciated the choice of this qualitative methodology for develop a tentative to a patient-centered conceptual model, as well as the specific method selected. I suggest to Authors, before Care Conceptual Model figure, to add a map: could be useful drawn a thematic map indicating patients main themes emerged. Also discussion part could be little bit expandend in the direction of results finding. It is not clear the statement "The participants of this study did not give written consent for their data to be shared publicly". Can better explain?

Reviewer 3 Report

Comments and Suggestions for Authors

The study is interesting.

The theoretical part is sufficiently well consolidated.

Qualitative methodology is logical.

I would have liked to see a combination of qualitative and quantitative methodologies.

The author could better qualify the novelty of the study.

Reviewer 4 Report

Comments and Suggestions for Authors

great and comprehensive work 

interesting as I did find the importance of developing this framework

I suggested some minor revisions to enhance your manuscript

your analysis is relevant 

and I encourage you to reinforce some parts of it what are unusual
